# Experimental Investigations on the Effect of Axial Homogenous Magnetic Fields on Propagating Vortex Flow in the Taylor–Couette System

**DOI:** 10.3390/ma12244027

**Published:** 2019-12-04

**Authors:** Thomas Ilzig, Katharina Stöckel, Stefan Odenbach

**Affiliations:** Chair of Magnetofluiddynamics, Measuring and Automation Technology, Dresden University of Technology, 01062 Dresden, Germany; katharina.stoeckel@tu-dresden.de (K.S.); stefan.odenbach@tu-dresden.de (S.O.)

**Keywords:** ferrofluid, Taylor–Couette flow, asymmetric boundary conditions, propagating vortices, magnetic fields

## Abstract

Experimental investigations of propagating vortex flow states (*p*V states) in a short Taylor–Couette system with asymmetric boundary conditions are presented. The flow state was established in a ferrofluid showing no magneto-viscous effect and was exposed to axial magnetic fields. It was found that the magnetic field led to a change in the spatial and temporal behavior of the *p*V state, indicating complex interactions between the flow field and magnetic field. A stepwise applied axial magnetic field destabilized the *p*V state, leading to an intermittent flow state. Gradually increasing the axial magnetic fields changed the temporal behavior of the regime. Up to magnetic field strengths of 20 kA/m, the orbital frequency, as a measure for the temporal periodicity, was increased with field strength.

## 1. Introduction

The Taylor–Couette system is a classical hydrodynamic model system, which has been in the focus of scientific interest for several decades now. Research has emphasized fundamental fluid dynamics, non-linear dynamics, self-organization and pattern formation etc., both experimentally and analytically [1,2,3,4,5,6,7]. Due to the fundamental mechanisms and principles, the effects known from the Taylor–Couette system are also of substantial importance for the understanding of flow and heat transfer phenomena. Taylor vortex structures also play a role in technical applications and can, for example, affect the overall lifetime of bearings or other fast-rotating machine parts.

In principle, the Taylor–Couette system benefits from a simple basic construction on the one hand and allows the investigation of various reliably reproducible flow states on the other. In its simplest form, the Taylor–Couette system consists of two coaxial cylinders of radii r_1_ and r_2_ and length L, which can rotate independently at angular velocities ω_1_ and ω_2_, respectively (Figure 1). A viscous fluid of kinematic viscosity ν fills the gap of the width d = r_1_ − r_2_ between the two cylinders. The radius ratio is given by η_RR_ = r_1_/r_2_ and the aspect ratio is given by Γ = L/d. Hence the Reynolds number of each cylinder can be calculated by:(1)Rei= ωi·di·riv, i = 1 or 2. 

In experimental Taylor–Couette systems, the liquid is enclosed by the axial end walls at the top and bottom, thus the reflectional symmetry of a theoretically infinitely long system is broken [8,9]. Depending on the main parameters Re_1_, Re_2_, η_RR_ and Γ, as well as on the boundary and initial conditions, various flow states can be established [3,8,9,10,11,12,13]. In general, two types of end walls have to be distinguished—rigid and rotating. Depending on the type of the end wall, the flow in the boundary layer at the end walls is influenced differently [14,15]. A rigid end wl leads to a radial inflow (Bödewadt boundary layer), while a rotating end wall leads to a radial outflow (Ekman boundary layer) [10]. Since most experimental investigations are carried out with symmetric boundary conditions (rigid end walls), only a few experimental setups with asymmetric boundary conditions are used [9,11]. Asymmetric boundary conditions break the mid-plane symmetry of the Taylor–Couette system, and thus the spatial symmetry of the flow states can be varied and anomalous modes can occur [16].

The conditions in the boundary layer are the driving force for local boundary vortices, the so-called Ekman cells [15]. The Ekman cells have a major influence on the behavior of the flow in the bulk region of the system and thus lead to various flow phenomena in the Taylor–Couette system [17]. The influence of the Ekman cells decreases exponentially with increasing distance from the end walls [17,18]. Hence the effect on the bulk region strongly depends on the aspect ratio Γ and thus in short Taylor–Couette systems the Ekman cells have a very strong influence on the bulk flow. This effect becomes less important in long systems but is never negligible [8,11,15,19].

The basic flow state in the Taylor–Couette system is the circular Couette flow (CCF), which occurs at low Reynolds numbers of the inner cylinder. The outer cylinder is at rest or slightly co- or counter-rotates. Since the forces acting on a fluid element, the centrifugal and the viscous forces, are in equilibrium, the fluid elements in the gap are moving on circular paths around the inner cylinder. At a certain critical Reynolds number, the CCF basis state becomes unstable and the centrifugal force prevails over the viscous force, thereby forcing the fluid elements on spiral paths around the inner cylinder. As a result, a first self-organized flow pattern occurs, the Taylor vortex flow (TVF), characterized by alternating in/up and out/downward flows arranged in vortex pairs. At higher Reynolds numbers, higher ordered instabilities occur, such as wavy vortex flows (WVF), which also show time dependence.

Also quite common in Taylor–Couette systems are propagating vortex states (*p*V). The most prominent representatives are spiral vortex flows (SPI) which occur as a first instability in co- as well as in counter-rotating Taylor–Couette flow. Propagating spirals are characterized by an open helical structure. The toroidal closed vortex structure and the rotational symmetry of a TVF are broken, thus the azimuthal wavenumber is M ≠ 0 [18].

By contrast, the here described *p*V state is characterized by a toroidal closed propagating vortex structure which corresponds to a “M = 0”-mode [10]. The onset of this flow phenomenon occurs as a secondary instability following a SPI state. This corresponds to the description which can be found in [10]. Other toroidal closed propagating flow states are described in [20] and occur as a primary instability. Note that propagating vortices with a toroidal closed structure are mostly known from ramped Taylor–Couette systems or systems stressed by external forces, such as pressure gradients or magnetic forces [11,21,22]. The here described propagating vortex state occurs in short Taylor–Couette systems (Γ_odd_ = 5 and Γ_even_ = 6) with asymmetric boundary conditions.

Ferrofluids are colloidal suspensions of magnetic nano-sized particles contained in a non-magnetic carrier liquid. The flow behavior of these fluids can be significantly affected by external magnetic fields. As a main parameter, the rotational viscosity of the ferrofluid may change under the influence of the applied field [23,24,25,26,27]. This change in viscosity is due to a hindrance of the free particle rotation when the fluid is subject to a shear flow under the influence of external magnetic fields [25,26,27]. In a flowing ferrofluid, the suspended particles are affected by a viscous torque, generated by the surrounding carrier liquid. If, additionally, an external magnetic field is applied, the magnetic moments of the particles align in the direction of the field and, in the case of magnetic hard particles, a magnetic torque is exerted on the particles. If the direction of the magnetic field and the vorticity of the fluid are not collinear, the magnetic torque counteracts the viscous torque, resulting in an increase in the apparent viscosity [25,26,27,28]. The hindrance of free particle rotation is considered to be the only effect for highly diluted suspensions, neglecting interparticle interactions [28]. In fluids with interparticle interaction, other effects may come additionally into account, known as the magneto-viscous effect [28]. Depending on their size and on their concentration, the particles not only align with the magnetic field but form structures, such as agglomerated clusters or chains [26]. These structure-induced effects also depend on the direction of the applied field, and might be considerably larger than the effects found in ferrofluids without the magneto-viscous effect.

From the above mentioned, it becomes clear that both the particle size D and the volumetric concentration Φ of the suspended particles, play an important role in the description of the effects concerning the viscous behavior of ferrofluids. Under the assumption of a highly diluted, monodisperse ferrofluid and a low magnetic field, Rosensweig [26] approached the Langevin law by a Taylor series expansion. The equation
(2)$$\chi_{\mathrm{in}}=\;\frac{\pi }{18}\cdot \Phi \cdot \mu_{0}\cdot \frac{D^{3}\cdot M_{0}^{2}}{k_{B}\cdot T}$$
containing the initial susceptibility χ_in_, the permeability of vacuum ε_0_, the domain magnetization M_0_, the Boltzmann constant k_B_ and the absolute temperature T can be solved for the particle diameter D, and gives a first approach to evaluate the particle size.

The volumetric concentration can be estimated by:(3)$$\Phi \;=\;\frac{M_{S}}{M_{0}}$$
where by M_S_ denotes the saturation magnetization [26].

However, the description of the non-equilibrium magnetization, e.g., in the case of ferrofluid flow, is very complex and cannot solely be covered by Equations (2) and (3). The dynamic behavior of ferrofluids is characterized by linking mechanisms of the magnetic field and the flow field. A well-established experimental approach to study these sophisticated interaction mechanisms is provided by investigations of ferrofluid flow in the Taylor–Couette system. Due to the variety of well-known flow states, one is allowed to conduct experiments in many manners. The topology of the Taylor–Couette system is characterized by stability and transition areas. The distinction between different flow regimes takes place on the stability limits in between these transition areas. It is already known from experimental as well as from numerical investigations that the stability limits and transition areas may be shifted when a ferrofluid flow is exposed to external magnetic fields [23,24,29,30,31,32,33,34,35,36,37]. Generally, the onsets of primary bifurcating flow states such as TVF or SPI states are shifted to higher Reynolds numbers with increasing field strengths, leading to a stabilization of the CCF basic flow state [29,31,32,36,38]. Reindl [37] described similar observations for secondary bifurcating flow states, namely the WVF. His investigations were carried out in ferrofluids showing an interparticle interaction and thus magnetic fields were capable of shifting the onset of wavy modes to significantly higher Reynolds numbers, which indicates a stabilizing effect on the primary TVF state.

Magnetic fields have a damping effect on the ferrofluid-flow in the Taylor–Couette system, which might affect the properties and characteristic key parameters of the above-mentioned flow states. Niklas [32] found that the wave number k depends on the applied field strength and direction. As the axial applied magnetic fields reduced the wave number, the radial applied fields increased k. Similar results were observed by Vislovic [32]. Reindl [35,36,37] proved the dependence of the wave number decreasing with increasing field strength experimentally for axial fields. He also observed a reduction in the flow amplitudes with increasing field strength. In time dependent flow states, magnetic fields might lead to a change in the temporal behavior; Reindl [35,36,37] found that the orbital frequency of the investigated WVF states increased with increasing field strength both in ferrofluids with particle interaction and without. Furthermore, he showed that the vortex propagation velocity increased with an increase in the applied field strength. Altmeyer et al. [29] found that, under the influence of transversal magnetic fields, the mode structure of wavy TVF states and wavy SPIs is changed.

In this work, experimental investigations of toroidal closed propagating vortices in a ferrofluid in a Taylor–Couette system are presented. Furthermore, the influence of external homogeneous axial magnetic fields on this flow regime was investigated.

## 2. Materials and Methods

### 2.1. Experimental Design and Flow Detection

The used Taylor–Couette system consisted of two coaxial cylinders with radii r_1_ = 10 mm for the inner cylinder and r_2_ = 20 mm for the outer cylinder. Hence the radius ratio η_RR_ resulted to 0.5. The gap width d = 10 mm and the system length L could be varied. Experiments were carried out in short systems, choosing lengths of L = 50 mm and L = 60 mm, leading to an aspect ratio Γ_odd_ = 5 and Γ_even_ = 6, respectively. Both cylinders could rotate independently and were driven by synchronous servomotors. Figure 2 shows a cross-sectional view of the used Taylor–Couette system and the shear cell.

The horizontal system boundaries were unsymmetrical. The top plate was rigid, hence a Bödewadt boundary layer was formed. The bottom plate was attached to the outer cylinder and thus rotated when the outer cylinder rotated. For this reason, an Ekman boundary layer was formed at the bottom plate.

The homogeneous magnetic field H was generated by two pairs of Helmholtz coils, which were combined in a Fanselau arrangement [39]. With a given size of the homogeneous field area of 200 mm × 200 mm × 200 mm, the coil dimensions in this arrangement were generally smaller than when using a simple pair of coils and, in addition, the homogeneity of the field of about 98% in the mentioned area was significantly better than in normal Helmholtz-arrangements. The field strength ranged from 0 kA/m to 45 kA/m.

Since ferrofluids are opaque, optical measuring techniques such as laser Doppler velocimetry could not be employed. For this reason, flow detection was carried out using ultrasound Doppler velocimetry (DOP2125, Signal Processing S.A., Savigny, Switzerland), which exploited the acoustic Doppler effect. In principle, this means that the detected ultrasound echo of a suspended particle in the fluid was shifted towards higher (lower) frequencies when the particle was moving towards (away from) the ultrasound transducer. The measured particle velocity was displayed as positive when the particle moved away from the transducer and vice versa. The ultrasound transducer was attached to the top plate and allowed for the time dependent measurement of the axial velocity component *w* over the whole system length. The orientation of the z-axis, which was the rotational axis of the experiment, was therefore upside down by convention. This setup was already used by Reindl [35,36,37], who showed that different flow regimes such as TVF, WVF and SPI were distinguishable due to their time dependent axial velocity profiles. The data were processed in MATLAB and the flow regimes were graphically visualized by space–time plots.

The current measuring setup of our experiment only allowed us to employ one ultrasound transducer at one azimuthal position at time. Thus, it was not possible to make assumptions on the basic structure of the here described *p*V state, only based on the ultrasound velocimetry measurements. In order to characterize the *p*V flow regime, pretests in silicone oil (Baysilone M10, Bayer AG, Leverkusen, Germany) were conducted. We added a volumetric fraction of 0.01% aluminium particles (d_mean_ = 0.045 mm) for flow visualization, which allowed us to intensively observe the regime’s basic structure. The findings of these pretests were extrapolated to the ferrofluid experiments by the Reynolds analogy. A video of such a visualization experiment is provided in the Appendix A section.

### 2.2. Characterization of the Used Fluids

The dynamic viscosity of the silicone oil–aluminium suspension was η = (10.79 ± 0.03) mPas and the suspension density was ρ = (936.92 ± 0.55) kg/m^3^, resulting in a kinematic viscosity of ν = (11.48 ± 0.64) mm^2^/s at a reference temperature of Tref = 20 °C.

The used ferrofluid in this paper was a magnetite-based ferrofluid in a kerosene carrier. The kinematic viscosity ν = (5.02 ± 0.18) mm^2^/s (Tref = 20 °C) was directly measured depending on the temperature with an Ubbelohde viscometer AVS350 (Schott Geräte KPG, Jena, Germany).

The magnetization curve of the fluid in Figure 3a was obtained by measurements in a vibrating sample magnetometer VSM (Lakeshore 7407, Westerville, OH, USA). From this curve, the saturation magnetization M_S_ = 28.4 kA/m and the initial susceptibility χ_in_ = 0.75 were determined. Using Equation (2) and M_0_ = 450 kA/m as a common value for the domain magnetization [28], the mean particle diameter D = 10.5 nm was found. As already mentioned, Equation (2) is only valid for monodisperse ferrofluids. However, this is a good first approach to estimate the particle size and evaluate the dominating relaxation process, which in this case is Néel relaxation. Therefore, the used fluid should not show any magneto-viscous effects. To further evaluate this assumption, the fluid was investigated employing a shear-rate-controlled cone-plate rheometer [40,41]. Within a homogeneous magnetic field ranging from 0 kA/m ≤ H ≤ 40 kA/m, no increase in viscosity with increasing magnetic field strength was observable and thus no structure formation effects were assumed nor was any other evidence of a magneto-viscous effect given. The according flow curves at the initial state (without field), at H = 40 kA/m and at the reference state (after the removal of the field) are presented in Figure 3b.

From Equation (3), we found a volumetric fraction of the contained magnetic material of Φ = 6.3%. Table 1 gives an overview over the main properties of the used ferrofluid.

## 3. Results and Discussion

### 3.1. General Discussion of the Investigated pV State

In order to characterize the experimental setup, first investigations in silicone oil were performed, evaluating the basic structure of the *p*V regime and the transition region between the SPI and the *p*V state.

These flow visualization experiments, according to the classic experiments by [1,3,4,5,9] etc., by addition of reflecting (aluminium) particles, clearly gave evidence of the closed toroidal structure of the here described flow regime, indicating axial symmetry and a “M = 0”-mode (see the uploaded video in the Appendix A section).

Figure 4a shows the time–space plot of the axial velocity component *w* of the SPI state established in the described Taylor–Couette system. The propagation started near the bottom plate of the system and moved upwards to the Ekman cell at the top lid, where the vortices were annihilated. This flow state occured in a counter-rotating Taylor–Couette system at Re_1_ = −173 and Re_2_ = 133.

The *p*V regime emerged out of the above-mentioned SPI state when the counter-rotation of the system was increased. Figure 4b allows the comparison of the SPI and the *p*V state. Note that Re_2_ was still 133, only Re_1_ was slightly decreased to −180, i.e., by about 3.5%, showing the extreme sensitivity of the flow states to the boundary conditions.

The propagation of the vortices started at the Ekman cell at the top lid of the system and moved downwards to the bottom plate. In the chosen system with a length of Γ = 5, a state of three vortex pairs was established. The first and the second pairs at the top and mid region of the system remained stationary, while the bottom pair propagated towards the gap. New vortices were generated at a defect area between the mid-pair and bottom pair, and the annihilation of the vortices took place at the bottom plate. Both the generation and annihilation were periodic in time, whereby the vortex annihilation preceded the vortex generation.

Although the Ekman cell remained stationary, an oscillating movement of the top and mid pairs down the gap was clearly visible. The vortices moved app. 5 mm down the gap. As a new vortex pair emerged out of the mid pair, it moved back towards the lid. This motion cycle was periodic but not uniform. While the downward propagation was slow and constant, the backward movement occurred suddenly. Hence, the propagation velocity was not constant.

In a direct comparison of both states we found that the phase propagation velocity of the SPIs was much higher than the vortex propagation velocity of the *p*V state, indicated by the inclination of the color-coded flow bands.

The shown flow states were established with an aspect ratio of Γ = 5. The odd aspect ratio was considered to have a stabilizing effect on the *p*V-state. However, a certain *p*V-state was also established in a system with Γ = 6. The space–time plot in Figure 5 shows clearly three vortex pairs: One stationary pair at the top plate, one oscillating pair in the mid region and one propagating pair in the lower bulk region. The generation of the vortices took place between pair 2 and 3, and the vortices were annihilated at the bottom plate. As known from the odd aspect ratio, the vortex generation and annihilation were periodic in time.

### 3.2. Influence of Axial Magnetic Fields

#### 3.2.1. General Aspects

As mentioned above, magnetic fields can strongly influence the flow states in the Taylor–Couette system. Since the odd aspect ratio favors a stable *p*V state, the following investigations with ferrofluids were undertaken in the system with Γ_odd_ = 5.

As a first example for the influence of a magnetic field on the *p*V state, Figure 6 shows an intermittent state which was achieved by establishing the *p*V state in a stable manner and applying stepwise a magnetic field H = 40 kA/m. We found a situation where the *p*V regime was temporarily disturbed by an overlaying SPI state. These overlaying SPIs occurred at field strengths above 20 kA/m. Both the *p*V state and the SPIs alternated unpredictably. Since the ferrofluid used showed no measurable change of viscosity in a magnetic field, this change of the flow state can be assumed to be a direct influence of the magnetic field on the flow structure, since it seemed not to be driven by a shift of the stability border by a viscous change.

#### 3.2.2. Influence on the Spatial and Temporal Behavior

While the appearance of the mentioned intermittent flow regime was forced by a stepwise change of magnetic field strength, gradual changes of field strength led to changes of the orbital frequency while the *p*V state itself remained stable. This was valid for magnetic fields up to at least 20 kA/m.

For an investigation of the respective changes, the magnetic field strength was increased in increments of 5 kA/m and the velocity profiles were acquired at each field strength for app. 7.5 min. At the end of each measuring cycle, the magnetic field was removed and the initial *p*V state was re-established. Figure 7a–c compares the *p*V state in the Taylor–Couette system with an aspect ratio of Γ = 5 at field strengths of 0 kA/m, 10 kA/m and 20 kA/m, respectively.

As shown, the temporal behavior of the *p*V state changes with increasing field strength. This can be seen directly by comparing the space–time plots of the three flow states. The *p*V state at H = 0 kA/m showed ten local extrema, while the *p*V state at 20 kA/m showed 14 local extrema within 150 s. This already shows the strong influence of axial magnetic fields on the temporal behavior of the regime.

For a quantitative analysis, the orbital frequency f_O_ was determined by calculating the FFT on the temporal domain. For this purpose, the temporal velocity profiles at a fixed axial position of app. 35 mm of three independent measuring cycles were analyzed. Figure 8 shows the f_O_ depending on the field strength H, normalized by the value of the initial flow state at H = 0 kA/m. It is clearly visible that the frequency increased with increasing field strength.

The observations made above show that a significant change in the temporal behavior appeared for magnetic field strengths up to 20 kA/m. Measurements at higher field strengths were also conducted, but due to a poor SNR during these measuring cycles, the obtained data were not reliably evaluable. This changed only at stepwise applied field strengths at which another dominant flow state appeared as seen in Figure 6 where the SNR was sufficient. As an example of the few utilizable measuring points in the field range between 20 kA/m and 40 kA/m, Figure 9 shows the *p*V state under the influence of an axial magnetic field of 25 kA/m. This state was part of the above described measuring cycles with a gradually increasing field strength. In this example, the normalized orbital frequency decreased rapidly. The value of f_O_ in this case reached app. 40% of the initial value. This single observation is of course not representative of the influence of magnetic fields on the *p*V state in general. More detailed experiments will have to be carried out for field strength regimes higher than 20 kA/m, which will be subject to future investigations.

Comparing the median averaged velocity amplitudes, Figure 10 shows the decreasing velocity amplitudes up to a depth of app. 35 mm with increasing field strength. Combined with the finding of an increase in the orbital frequency, as shown above, this leads to the conclusion that the axial upward velocity overall has increased.

Below this depth, the amplitude maxima increase with the field, indicating that the axial downward velocity increased. This finding is in agreement with the findings of Reindl for the WVF and SPI states in magnetic fields [37].

The individual measuring point at 25 kA/m will also be discussed as an example and its velocity amplitudes are shown in Figure 10 (bottom right). Compared to the initial flow state without a magnetic field, the sinusoidal velocity profiles at H = 25 kA/m are shifted by about half a period upward. This indicates a change of the axial wave number k by a reduction of the number of vortices contained in the flow. It is known from TVF states exposed to magnetic fields that increasing field strengths lead to lower axial wave numbers [35]. Due to the time-dependent nature of the here described *p*V states, results from the time-independent TVF states are not directly transferable. However, the made observations may indicate that the radial in-flow/axial up-flow at the rigid top plate is diminished or even disappeared, leading to a flow state with an uneven number of vortices.

The presented results show a complex interaction between the flow field and the magnetic field. Since a ferrofluid with no particle–particle interaction was used and, furthermore, no measurable effect on the rotational viscosity was found, all described effects are considered due to a direct influence of the magnetic field on the flow structure. This linking between flow and magnetic field is not fully understood and further investigations will be carried out with special focus on these interaction processes.

## 4. Conclusions

In this work, a special type of propagating vortex flow is presented. This flow state was established in a Taylor–Couette system with a short aspect ratio and asymmetric boundary conditions. While the top plate of the shear cell is rigid, the bottom plate rotates with the outer cylinder. As known from the literature, the short aspect ratio is considered to intensify the influence of the Ekman cells on the bulk region.

It has been shown that the investigated flow regime emerges out of primary bifurcating SPIs in a counter-rotating Taylor–Couette flow, when the counter-rotation is increased. The described *p*V state is characterized by a complex spatial and temporal behavior of oscillating and propagating toroidal closed vortices. The propagation is asymmetric from top to bottom, which is considered to be due to the boundary conditions. Furthermore, the propagation is periodic in time.

The *p*V state was established in a ferrofluid and exposed to external homogeneous axial magnetic fields of varying field strengths. The flow state was found to be very sensitive to these magnetic fields and could be altered significantly in its temporal and spatial behavior. The main observations in this paper are as follows:A strong stepwise applied magnetic field disturbed the *p*V state in such a manner that the flow was temporarily superimposed by SPIs.The orbital frequency increased by 40% with increasing field strengths up to 20 kA/m when the field strength was applied and increased gradually. Furthermore, the vortex propagation velocity increased with increasing field strengths.At field strengths of above 20 kA/m, the data basis during the conducted measuring campaigns was not reliable due to insufficient SNR. However, some of the made observations indicate that major changes in the temporal and the spatial behavior of the *p*V state may occur at higher field strength regimes. Without claiming to make generally valid statements, the change in the wavenumber k and the strong decrease in the orbital frequency f_O_ to app. 40% of the initial value should be mentioned as examples.

The used ferrofluid showed no evidence of a magneto-viscous effect or a measurable increase in the viscosity in a cone-plate rheometer under the influence of an external magnetic field. Thus, the described results are associated with neither structure formation processes nor an increase in the rotational viscosity respectively. The more remarkable are the results presented in this paper, considered to be due to a linking of the flow field and the magnetic field.

The full parameter range in which the *p*V state is stable is not known yet. Further experimental and numerical studies will have to be undertaken to investigate the topology of these flow states in ferrofluids in the presence of magnetic fields.

## Figures and Tables

**Figure 1 materials-12-04027-f001:**
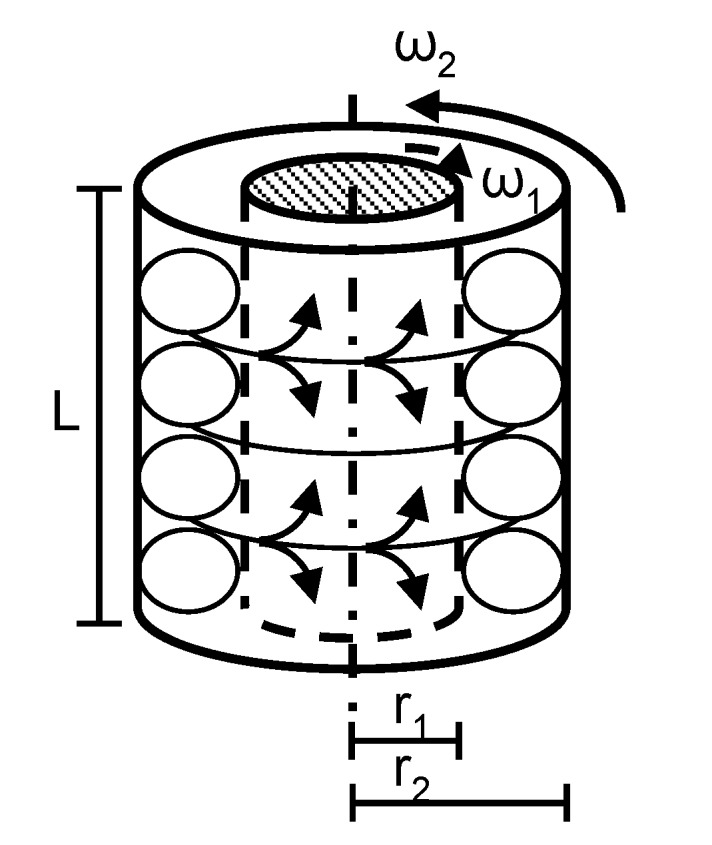
Principle sketch of the Taylor–Couette system. The main components are the inner cylinder with radius r_1_, rotating at angular velocity ω_1_; the outer cylinder with radius r_2_, rotating at angular velocity ω_2_. The fluid cell has the system length L.

**Figure 2 materials-12-04027-f002:**
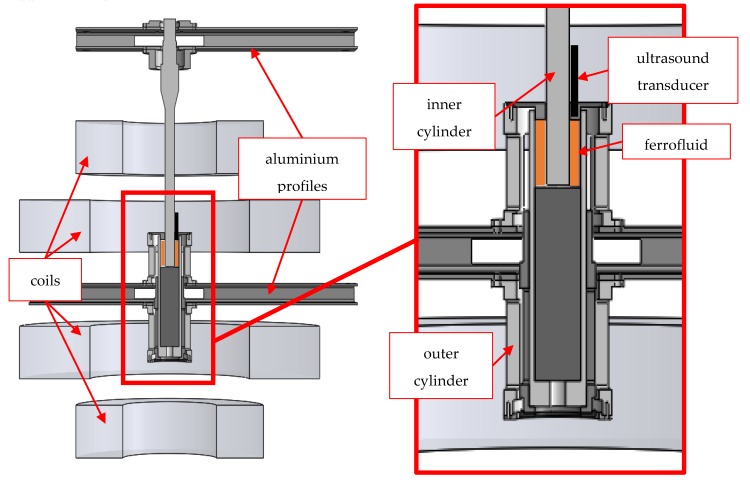
Cross-sectional view of the Taylor–Couette experiment and a detailed view of the shear cell.

**Figure 3 materials-12-04027-f003:**
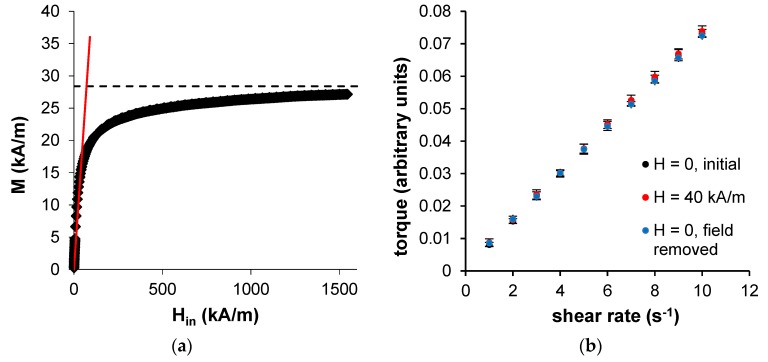
(**a**) Experimental magnetization curve of the used ferrofluid. The data were acquired employing a vibrating sample magnetometer. Saturation magnetization M_S_ = 28.4 kA/m (black dotted line); initial susceptibility χ_in_ = 0.75 (inclination of the red solid line). (**b**) Flow curves of the ferrofluid obtained by a shear-rate-controlled rheometer. The linear relation between the shear rate and the measured torque indicates no field dependence of the viscosity nor any structure formation or other magneto-viscous effects.

**Figure 4 materials-12-04027-f004:**
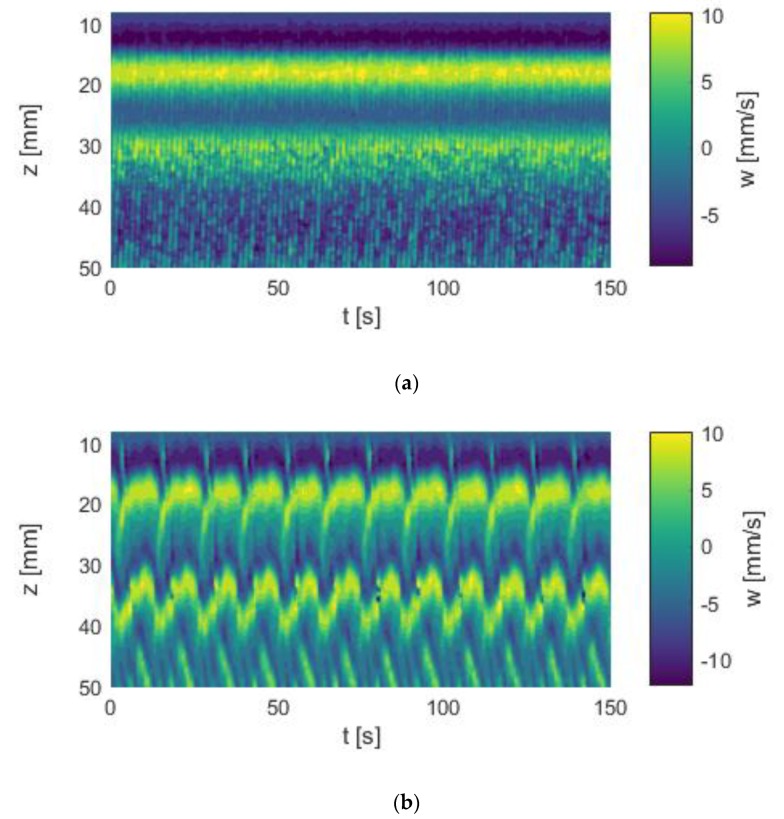
Space–time plots of axial velocity profiles. (**a**) SPI flow state, Re_1_ = −173, Re_2_ = 133. (**b**) *p*V flow state, Re_1_ = −180, Re_2_ = 133. Both measurements were carried out in silicone oil in a short Taylor–Couette system with Γ_odd_ = 5.

**Figure 5 materials-12-04027-f005:**
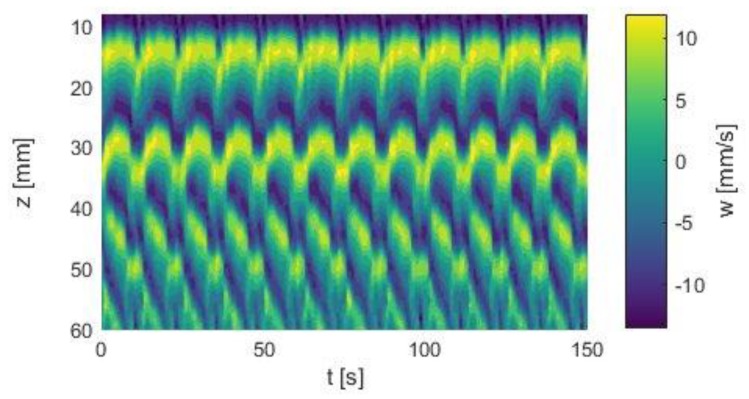
Space–time plots of axial velocity profiles of a *p*V flow state. Re_1_ = −188, Re_2_ = 122. The measurements were carried out in silicone oil in a short Taylor–Couette system with Γ_even_ = 6.

**Figure 6 materials-12-04027-f006:**
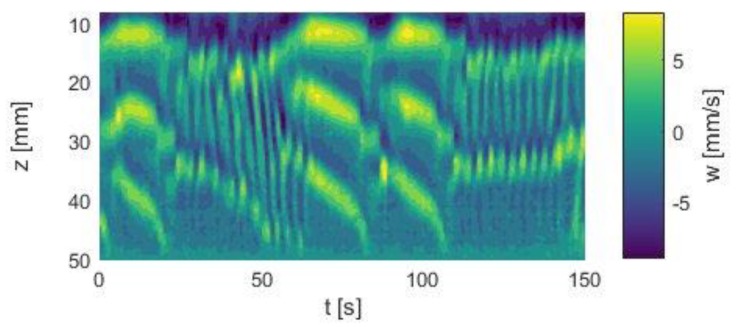
Intermittent flow regime in the Taylor–Couette system with an aspect ratio of Γ = 5. The basic *p*V state is irregularly disturbed by overlaying SPIs. Re_1_ = −185, Re_2_ = 135, H = 40 kA/m.

**Figure 7 materials-12-04027-f007:**
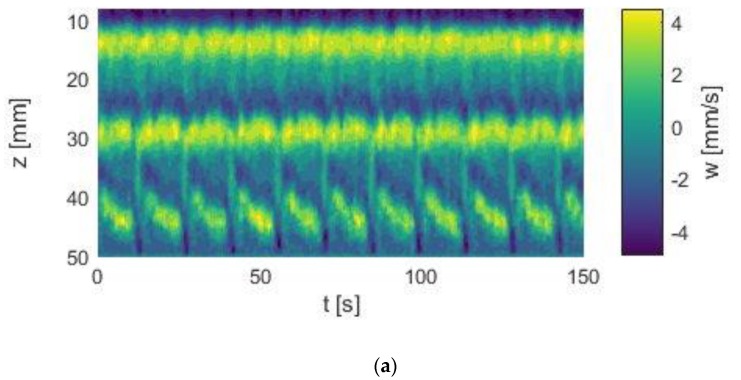
Comparison between (**a**) the initial *p*V state at 0 kA/m and the field affected states at (**b**) 10 kA/m and at (**c**) 20 kA/m. Re_1_ = −175, Re_2_ = 130. The number of local maxima in the space–time plots increases with increasing field strength, indicating that the orbital frequency changes with the field.

**Figure 8 materials-12-04027-f008:**
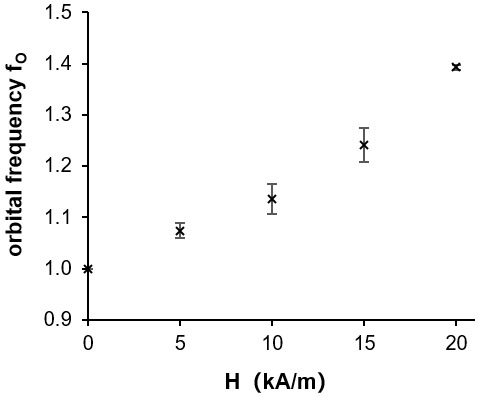
Orbital frequency f_O_ depending on magnetic field strength H. With increasing field strength, the frequency increases.

**Figure 9 materials-12-04027-f009:**
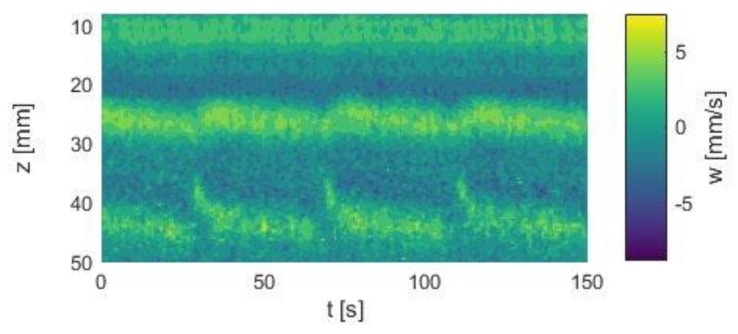
The *p*V state exposed to an axial magnetic field of 25 kA/m. This data point was part of a measuring cycle of gradually increasing field strength. Due to a poor SNR at field strengths above 20 kA/m during these cycles, the measured data were not reliably evaluable, so that this plot represents one of the few exemplary measuring points for higher field strengths. An FFT analysis showed a significant decrease in the normalized orbital frequency f_O_ of 0.42. Re_1_ = −175, Re_2_ = 130.

**Figure 10 materials-12-04027-f010:**
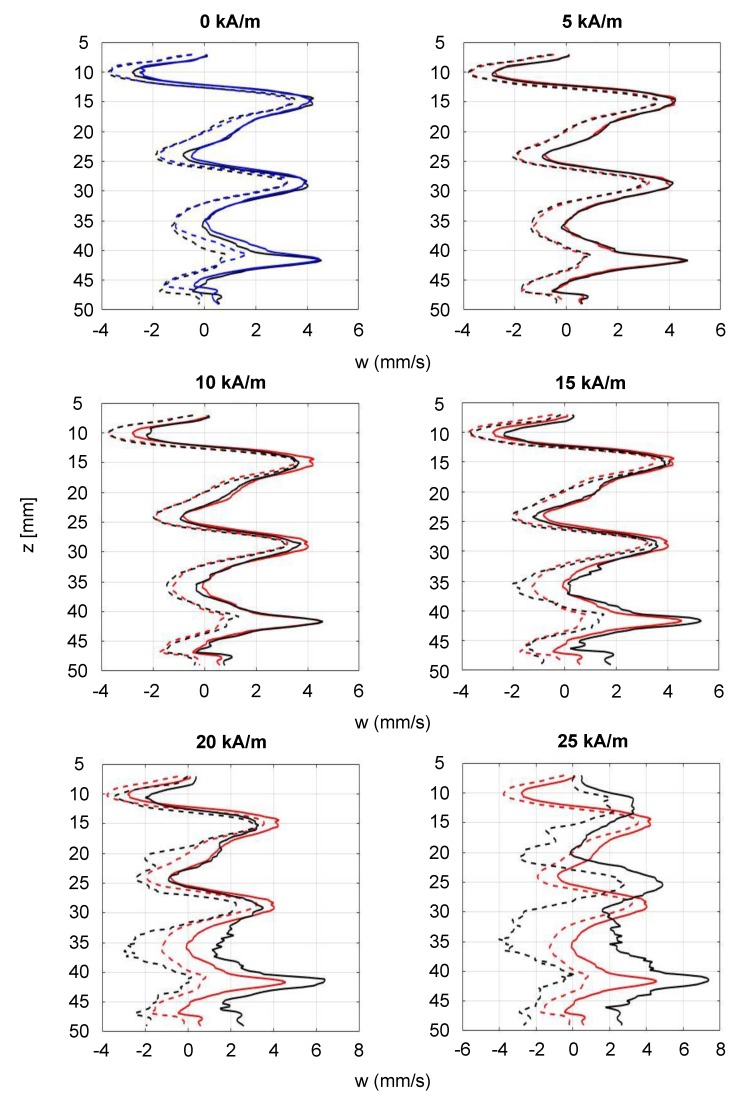
Median averaged velocity profiles of the *p*V state, depending on the magnetic field strength. The dotted lines show amplitude minima, the solid lines amplitude maxima. The black curves show the amplitudes at given field strengths, the red curves compare them with the initial amplitudes at H = 0 kA/m. The blue curve in the top left plot shows the amplitudes after the measurements when the magnetic field was removed.

**Table 1 materials-12-04027-t001:** Basic properties of the ferrofluid used in the experiments.

Property	Value	Unit
Magnetic material	Magnetite	-
Volumetric concentration (Φ)	0.063	-
Saturation magnetization (M_S_)	28.4	kA/m
Initial susceptibility (χ_in_)	0.75	-
Kinematic viscosity (ν)	(5.02 ± 0.18) × 10^−6^	m^2^/s

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
