# Peer review of "Experimental Investigations on the Effect of Axial Homogenous Magnetic Fields on Propagating Vortex Flow in the Taylor–Couette System"

_materials, 2019, doi:10.3390/ma12244027_

Round 1
Reviewer 1 Report
The authors present in this work experimental investigations on the effect of axial homogenous magnetic fields on propagating vortex flow in the Taylor-Couette system. The quality of the work is high enough and appropriate to be published in the journal Materials, but I suggest to gather some more data points and more data analysis.
The Introduction part clearly and most properly described the statement of the study, literature reviews, and what has been done in this field of research. It would be even nicer if, in one paragraph, describe the importance of this study and even explain which industries might be beneficial as a result of this study ( like the Heat transfer industry).
Also, the method of the velocity measurement has been referenced to the dissertation, but it would be nice to describe it in this paper ( A paragraph) briefly.
What is the reason for using two pairs of Helmholz coils instead of one?
The author several times mentioned that the ferrofluid used in this experiment did not experience any change in viscosity as a result of the applied external magnetic field of 0 to 40 KA/m, yet it is suggested to bring experimental proof for this statement.
In fig 8 to describe the behavior of the ferrofluid after 20 KA/m, I strongly recommend getting at least two more data points to be able to describe the behavior more reliably. One data point is not enough to show behavior.
Fig 9 looks very busy. As a suggestion, it would be nice to rotate the figures by -90 or some changes to be more prominent.
Also, do you think the applied magnetic field is rotating the vortex's axial?
This paper includes fascinating observations and a sufficient background review. It would be nice to have more work on data analysis, which looks a bit inexpressive.
Reviewer 2 Report
This is an interesting and clearly written paper on experimental work in Taylor-Couette flows using a ferrofluid. The interaction between the imposed magnetic field and the ferrofluid particles has a clear effect on the observed flow structures, and the results would definitely be of interest to other workers in either Taylor-Couette flows or ferrofluids. The work is therefore clearly publishable in Materials. I only have one comment that definitely must be addressed, plus one other suggestion that is optional.
Regarding the point that must be addressed, on page 2 it states "By contrast, the here described pV state ... which corresponds to a "M=0" mode." That is, you are asserting that your solutions here are axisymmetric, is that correct? However, later on, when you describe your experimental apparatus, there is only mention of one ultrasound transducer. If the flow is only being measured at one location in azimuth, surely you cannot possibly say though whether it is axisymmetric or not?? So either you do actually have multiple transducers, in which case you should carefully describe how many and where they are placed, or else you can't be sure that your measured signal is actually axisymmetric. (The results presented here would still be interesting even if you cannot be sure whether the signal is axisymmetric or not, but this issue should definitely be clarified.)
The optional point concerns some of the references. The very first sentence lists references 1-5 as presumably being general background to Taylor-Couette flows, but none of these five is the kind of review paper I was expecting for general background.. Also, still on the subject of references, I believe that reference 34 is actually about magnetohydrodynamic Taylor-Couette flow rather than ferrofluid Taylor-Couette flow? And if you decided that you do want to contrast ferrofluids with MHD (they both involve externally imposed magnetic fields, but are otherwise very different!), then again a review paper (such as Rudiger et al, Physics Reports 741, 1-89, 2018) would probably be better than just some otherwise random (?) paper on MHD Taylor-Couette flows.
Round 2
Reviewer 1 Report
Dear Authors,
It is a decent job.
Just in figure 10, when you rotate the figure, I would suggest turning back the numbers.
Each figure caption (..KA/m) on top, speed to the left, and rotate z(mm) upside down.
Regards,
Author Response
Dear reviewer,
I am sorry that I misunderstood your intention in your first review. Now I got your point, at least I think so.
I would suggest the following: I re-arranged figure 10 so that it is now a 2 x 3 subplot (instead of a 3 x 2 subplot before), but I keep the orientation of the individual subplots. This is because in this orientation the displayed length scale (z-axis) corresponds perfectly to the orientation of the rotational axis in the Taylor-Couette experiment. Further, the ultrasonic transducer is attached to the rigid top plate and measures the velocity profiles pointing downwards in the gap. Our coordinate system is therefore upside down. So I in this arrangement the plots seem to me more intuitively to read.
I furthermore will add a brief comment on that in the papers manuscript in the materials and methods section on p. 4.
I hope you'll agree to that.
Kind Regards.